# Development of an Event-Specific Droplet Digital PCR Assay for Quantification and Evaluation of the Transgene DNAs in Trace Samples of GM *PRNP*-Knockout Goat

**DOI:** 10.3390/foods11060868

**Published:** 2022-03-18

**Authors:** Wenting Xu, Ping Shen, Rong Li, Biao Liu, Litao Yang

**Affiliations:** 1Joint International Research Laboratory of Metabolic and Developmental Sciences, School of Life Sciences and Biotechnology, Shanghai Jiao Tong University, Shanghai 200240, China; wenting.xu@sjtu.edu.cn (W.X.); lirong190805020@163.com (R.L.); 2Development Center of Science and Technology, Ministry of Agriculture of People’s Republic of China, Beijing 100025, China; shenping84@hotmail.com; 3Key Laboratory on Biosafety of Nanjing Institute of Environmental Sciences, Ministry of Ecology and Environment, Nanjing 210042, China; liubiao@nies.org

**Keywords:** ddPCR, genetically modified *PRNP*-knockout goat, event-specific, trace samples

## Abstract

The prion protein (*PRNP*) gene encoding prion protein is considered a prerequisite for the occurrence of scrapie disease, and knockout of the *PRNP* gene in transgenic goat is one effective approach to avoid scrapie. This study aims to establish an event-specific droplet digital polymerase chain reaction (ddPCR) assay to detect and quantify the content of genetically modified (GM) *PRNP*-knockout goat event KoP1. The developed ddPCR assay presents high specificity, sensitivity, accuracy, precision and wide dynamic range. The limits of detection and quantification were as low as 1.44 and 7.2 haploid genome equivalent (HGE) per reaction, respectively. Furthermore, this assay was successfully applied in quantifying the goat KoP1 GM content in milk, feces and living environmental soil samples. We believe that the developed ddPCR assay has the potential to be used in the evaluation of horizontal gene transfer and the practical risk assessment of GM goat event KoP1 and its derivatives.

## 1. Introduction

As the recombinant DNA techniques continue to improve, more and more transgenic animals have been developed for scientific research and approved for commercialization. In 2002, a transgenic pig with 1,3-galactosyltransferase gene knockout was reported which can be used in organ xenotransplantation research [1]. The *PrP* gene which encodes prion protein (*PRNP*) was knocked out to produce homozygous GM cattle or goat fetuses via a sequential gene targeting system [2]. In 2015, a transgenic salmon harboring a growth hormone gene was approved for use in food materials [3]. Although transgenic animals have been produced and commercialized for quite some times, the public still has concerns about the safety issues of transgenic animals and their products, with particular emphasis on risks of the escape (release) of transgenic animals and their impact on the environment, horizontal gene transfer, and food safety, etc. [4]. Thus, most countries and regions around the world have issued a series of laws and regulations to regulate the research and application of genetically modified animals [5]. For example, the developers of genetically modified organisms (GMOs) were usually requested to supply the full data of risk assessment before applying for authorization of GM food/feed, including the proper GM event-specific detection and quantification method [6]. Therefore, the development of sensitive detection and quantification methods for transgenes is essential for the implementation of safety assessment and GM labelling regulation.

To establish effective detection methods for transgenic animals and their products, techniques targeting exogenous DNAs and proteins are often used, such as conventional polymerase chain reaction (PCR), quantitative real-time PCR (qPCR), enzyme-linked immunosorbent assay (ELISA), and lateral flow strip (LFS), etc. [7]. The DNA detection methods for GMOs are grouped into three categories according to their specificities to exogenous DNA sequences: element-specific, construct-specific, and event-specific [8,9]. Normally, element-specific methods target the commonly used transgenic elements (*CaMV35s* promoter, *NOS* terminator, and *NptII* gene, etc.) while event-specific methods detect the junction region between recipient genome and exogenous inserted DNAs. They are mainly used in routine lab analysis of GMOs. Screening methods are firstly applied to preliminarily identify whether or not the samples contain GM components, then an event-specific method is used to further identify and quantify the special GM event when the screening test result is positive [10,11]. The golden standard for DNA target quantification in many research fields is qPCR employing TaqMan chemistry, which has been widely applied in clinical diagnosis, foodborne pathogen detection, and GM content detection, etc. For example, one gene-specific qPCR method targeting Atlantic salmon (*Salmo salar* L.) growth hormone gene (*GH1*) was developed to quantify the *S. salar* L. DNA and Salmonidae ingredients in commercial foods [12]. We previously reported an event-specific qPCR method for quantifying the GM content of GM *hLZ* goat event [13]. However, there is still a need to construct the standard curves with series of calibrators for that qPCR analysis. Digital PCR (dPCR) is a new DNA quantification technique which can use the same primer/probe set developed for qPCR. In dPCR, the DNA template and PCR mixture is divided into thousands of individual partitions, and each partition acts as a separate qualitative test with a digital (positive or negative) outcome, regardless of the amplitude of the signal. By counting the negative and positive reads, the number of template DNA copies in a sample is calculated directly using Poisson distribution [14]. dPCR offers several advantages over qPCR, such as high sensitivity, absolute quantification of DNA without the need of calibrators, and high resistance to PCR inhibitors, etc. [14]. dPCR has been accepted and applied in many aspects of GMO analysis, including copy number determination, quantification of trace GM content samples, and multiplex quantification, etc. [15]. For example, Collier et al. used droplet digital PCR (ddPCR) to evaluate the transgene copy numbers in various crops, including rice, citrus, potato, maize, tomato, and wheat [16]. Košir et al. confirmed that ddPCR is an effective tool for GM soybean GTS 40-3-2 content quantification in processed feeds by comparing the results between ddPCR and qPCR quantification [17]. Scollo et al. used ddPCR to quantify olive oil DNA and to evaluate various oil DNA extractions and amplification methodologies [18]. Deconinck et al. developed one species-specific ddPCR assay for semi-quantitative evaluation of the Atlantic salmon content in processed food products [19]. Köppel et al. proved that compared to qPCR, the ddPCR approach has higher efficiency when quantifying samples containing trace GM contents [20]. To date, only a few ddPCR assays for GM animal content in risk assessment have been reported [19,21]. 

Sheep pruritus is an infectious and incurable prion disease which is caused by variant prion protein (*PRNP*) [22]. Knockout of animal endogenous prion protein gene could make animals resistant to this disease, and is of great significance to public health and agricultural economy. Transgenic *PRPN* knockout goat event KoP1 was developed to generate GM goat lines which are resistant to prion disease by the Shanghai Transgenic Research Centre, China, and is currently in the pipeline of application for commercial license [23]. In this study, we developed a KoP1 event-specific ddPCR assay with high specificity and sensitivity, and successfully applied this method to quantify trace KoP1 GM contents in milk, fecal, and living environment samples for risk assessment.

## 2. Materials and Methods

### 2.1. Preparation of Test Materials

Transgenic *PRNP* knockout goat event KoP1 and non-GM (NGM) goats were developed by the Shanghai Transgenic Research Centre, China [23]. The goat milk, feces, and living environmental soil samples were supplied by Shanghai JieLong Biotech Company, China. Goat arterial blood samples (10 mL each) were taken from the carotid artery with a disposable syringe and quickly placed into blood collection tubes coated with anticoagulant. For performance evaluation of the ddPCR assay, four DNA samples coded with B1–B4 were prepared by mixing the GM KoP1 genomic DNAs with that of NGM goat, and the expected GM DNA percentages were 1.7%, 0.85%, 0.50%, and 0.25% for B1–B4 samples, respectively. Four milk samples (M1–M4) were collected from individual GM KoP1 female goat. A total of 18 feces samples (coded from F1 to F18) from 16 GM KoP1 and two NGM goats were collected shortly after goats discharged. Five soil samples (S1–S5) were sampled from the environment where GM KoP1 goat feces were deposited. Blood samples from GM human *lysozyme* (*hLZ*), GM human *lactoferrin (hLF)*, and GM human *serum albumin (hSA)* goats were used for the specificity test, which was kindly supplied by Shanghai JieLong Biotech Company, China. The detail of milk, feces, and soil samples are described in Appendix A. All animals were taken care of in accordance with the conditions of the Association for the Assessment and Certification of Laboratory Animal Care (AAALAC). 

### 2.2. DNA Extraction and Purification

Genomic DNA was isolated using commercial DNA extraction kits according to the manuals. Blood genomic DNA was extracted and purified from 200 µL blood samples using TIANamp Genomic DNA Kit (DP304, TIANGEN Biotech, Beijing, China). Fecal DNAs were extracted from 500 mg fecal samples using QIAamp Fast DNA Stool Mini Kit (51604, QIAGEN China (Shanghai) Co.,Ltd. Shanghai, China). The soil DNAs were extracted from 500 mg soil samples using DNeasy Power Soil Kit (142579, QIAGEN). The amounts of isolated DNAs were measured and evaluated using a Nanodrop 2000 (Thermo Fisher Scientific, Waltham, MA, USA). The quality of purified DNA was also evaluated by 1% agarose gel electrophoresis (120 V, 25 min) in 0.5 × TBE with GelRed staining. All the extracted DNAs were stored at −20 °C before using.

### 2.3. Oligonucleotide Primers and Probes

The primers and TaqMan probe were designed targeting the event-specific gene sequence with Beacon designer software version 8.0 (PREMIER Bio soft, San Francisco, CA, USA). The expected amplicon length of the event-specific DNA fragment is 131 bp. The locations and sequences of event-specific primers (GM-Prion-F/-R) and probe (GM-Prion-P) are shown in Figure 1. Goat *prolactin receptor* (PRLR, GenBank Accession Number AF041979.1) gene was used as species reference gene, and the primers and probe were previously designed in our lab [24]. Sequences of primers and probes are listed in Table 1. All primers and probes and were purchased from Thermo Fisher Ltd. (Shanghai, China).

### 2.4. Conventional PCR

Conventional PCR reaction was prepared with a final volume of 25 µL, which contained 12.5 µL Taq PCR Master Mix (Huirui Company, Shanghai, China), 1 µL forward primer, 1 µL reverse primer (10 µM), 2 ng DNA template, and 8.5 µL of ddH_2_O. The PCR was performed on a Veriti certified refurbished thermal cycler (Thermo Fisher Ltd., Shanghai, China) with the program: 5 min at 95 °C, 35 cycles of 95 °C for 10 s, 60 °C for 30 s, and 72 °C for 1 min, plus an additional step at 72 °C for 7 min. PCR products were analyzed by electrophoresis in 2.0% (*w*/*v*) agarose gels with GelRed staining.

### 2.5. ddPCR

The ddPCR was performed on a QX200 Droplet PCR platform (Bio-Rad, Pleasanton, CA, USA). ddPCR reaction was prepared with a final volume of 20 µL, which was comprised of 10 µL of 2× ddPCR Supermix (Bio-Rad, Pleasanton, CA, USA), 1 µL forward primer (10 μM), 1 µL reverse primer (10 μM), 0.4 µL probe (10 μM), 1 µL DNA template, and 6.6 µL ddH_2_O. After the 20 μL reaction mix was prepared, the mixture was transferred to 8-well cartridges for emulsion droplets generation using a QX200 droplet generator (Bio-Rad, Pleasanton, CA, USA). Then, the generated water-in-oil droplets were transferred to a 96-well plate for amplification with a T100 PCR cycler (Bio-Rad, Pleasanton, CA, USA). The program for ddPCR amplification was: 95 °C for 5 min; 40 cycles of 30 s at 95 °C and 60 s at 58 °C. After thermal cycling, 98 °C for 10 min and then cooled to 4 °C. After PCR amplification, the 96-well plate was transferred to a QX200 droplet reader (Bio-Rad, Pleasanton, CA, USA) for fluorescent signal monitoring. The readout data was analyzed using QuantaSoft (Bio-Rad, Pleasanton, CA, USA). 

### 2.6. Performance Evaluation of the Event-Specific ddPCR Assay

In order to evaluate the event-specific ddPCR assay for GM goat event KoP1, key parameters including specificity, sensitivity, dynamic range, repeatability, accuracy, and precision were assessed carefully. Specificity was tested employing genomics DNAs from various GM goat lines (GM *hLA*, GM *hSA*, and GM *hLF*) and NGM goat line, as well as ddH_2_O. To evaluate the sensitivity, seven serial diluted genomic DNAs of GM event KoP1 with concentrations corresponding to 4500, 900, 180, 36, 7.2, 1.44, and 0.29 haploid genome equivalents (HGE)/µL were prepared and used. In dynamic range and repeatability evaluation, seven serial diluted genomic DNAs of GM event KoP1 with concentrations corresponding to 16,000, 1600, 160, 50, 25, and 10 HGE/µL were tested. For evaluating the precision and accuracy of the ddPCR assay, four samples (coded S1–S4) with known GM KoP1 contents were quantified. Each sample was repeated three times, and each time with three parallel reactions.

### 2.7. ddPCR Analysis of Practical Samples

The evaluated KoP1 event-specific ddPCR assay was then used to measure GM contents in practical samples. Specifically, four milk samples, 18 fecal samples, and five soil samples from the goat living environment were tested. Each sample was tested three times.

## 3. Results and Discussion

### 3.1. The Event-Specific ddPCR Assay of GM Goat Event KoP1 Has High Specificity

Before evaluating the specificity of the developed ddPCR assay, the specificity of designed primers was firstly tested using conventional PCR. In the conventional PCR analysis, a DNA fragment with the expected length of 131 bp was only amplified and observed in the reactions using DNA template of GM event KoP1, and no amplicon was obtained in reactions using DNAs of other GM goat lines (GM *rhSA* line, GM *rhLZ* line, GM *rhLF* goat line), NGM goat, and ddH_2_O (no template control, NTC) (Figure 2a). 

In the KoP1 event-specific ddPCR assay, the specificity was evaluated using different GM goat lines and non-GM goat line as DNA templates. Positive droplets and fluorescent signals were only observed in the reactions containing GM event KoP1 DNA, and no positive droplet and fluorescent signal was obtained in the reactions containing DNAs from other goat lines and NTC (Figure 2b). Thus, both the conventional qualitative PCR and ddPCR results showed that the established event-specific ddPCR assay has high specificity for the KoP1 event.

### 3.2. KoP1 ddPCR Assay Presents Higher Sensitivity than qPCR

In general, limit of detection (LOD) and limit of quantification (LOQ) are tested to evaluate the sensitivity of a new quantitative analytical method. LOD is defined as the lowest template amount that can be detected reliably at the 95% confidence level. LOQ is defined as the lowest template amount that can be quantified with high accuracy and good coefficient of variation (CV). For quantitative GMO analysis, the quantitative bias and CV should be less than 25% [25]. In the KoP1 ddPCR assay, fluorescent signals and positive droplets were observed and quantified in the reactions containing 4500, 900, 180, 36, 7.2, and 1.44 copies of HGE KoP1 DNA, but not in the reaction with 0.29 copies HGE DNA (Table 2), indicating that the LOD of the KoP1 ddPCR assay was 1.44 HGE per reaction. In the LOD test, the absolute amount of template DNA in each positive reaction was quantified. The relative standard deviation (RSD) values ranged from 4.15% to 16.58, the quantified bias values were less than 25% when template DNA amount was more than 1.44 copies HGE (Table 2). Therefore, the LOQ was determined to be 7.2 copies HGE per reaction. The LOD and LOQ of ddPCR assay was lower than those of qPCR assays [7,12,13], indicating that the established KoP1 ddPCR assay is highly sensitive and has the potential to be applied in quantifying the KoP1 GM content in trace samples.

### 3.3. The KoP1 ddPCR Assay Has Wide Dynamic Range and Good Repeatability

A reliable and effective dynamic range from LOQ to the highest concentration of template that could be quantified is crucial for a precise quantitative digital PCR assay. A total of nine blood genomic DNA dilutions with concentrations of 16,000, 1600, 160, 50, 25, 10, 7.2, 5, and 1 HGE/μL were used to evaluate the dynamic range of the KoP1 ddPCR assay. Except for assays containing 5 and 1 HGE template DNA per reaction, all of the other diluted DNAs were accurately quantified with the bias less than 25% (Table 3). The linear curve was drawn between the quantified and expected amounts of template DNA (Figure 3). A linearity (R^2^) of 1.000 was obtained within the dynamic range of 16,000–7.2 copies HGE per reaction, which was comparable with the requirement of qPCR and ddPCR [25,26,27]. 

Standard deviation (SD) and relative standard deviation (RSD) were calculated to evaluate the repeatability of the KoP1 ddPCR assay. The RSD values for all reactions within the dynamic range were from 3.64% to 21.57% (Table 3), showing that the KoP1 ddPCR assay has good repeatability according to the Minimum Information for Publication of Quantitative Digital PCR Experiments guidelines [26,27].

### 3.4. KoP1 ddPCR Quantification Results Were Accurate in Simulated Samples 

Four simulated goat blood samples (B1–B4) containing known KoP1 GM content at 1.7%, 0.85%, 0.5%, and 0.25% were quantified using the ddPCR assays of KoP1 event and goat endogenous reference gene *PRLR*, and the results were expressed as the copy number ratio of KoP1 event to *PRLR*. As shown in Table 4, the detected GM contents of samples S1–S4 by the ddPCR were 1.71%, 0.86%, 0.55%, and 0.28%, respectively. The quantified bias between the true values and quantified GM contents were within the range of 0.59%–12.00%, which were far below the acceptable criterion of 25% [25]. These results confirmed that the KoP1 ddPCR assay is accurate and suitable for quantification of samples with various GM KoP1 contents, even for sample with KoP1 GM content as low as 0.25%.

### 3.5. KoP1 ddPCR Assay Was Successfully Applied to Practical Trace Samples Analysis 

Since the developed ddPCR assay showed high sensitivity and good performance in simulated samples, it was further tested in assessing KoP1 GM content in practical trace samples for risk assessment. Three types of practical trace samples were collected and tested, including four goat milk samples (M1–M4), 18 goat fecal samples (F1–F18), and five living environmental soil samples (S1–S5). All samples were tested using the ddPCR assay in three parallel reactions, and the average quantified GM amounts, SD, and RSD values are shown in Table 5. 

In the goat milk samples, the absolute GM amount of KoP1 event was detected and quantified in all four milk samples (S1–S4). The quantified amounts of GM KoP1 were 1346.7, 1020.0, 919.3, and 832.7 HGE per reaction in S1–S4, respectively. The RSD of detected KoP1 amounts in S1–S4 samples ranged from 0.86% to 3.53%. These results showed the general existence of genomic DNAs in goat milk samples, which may be derived from somatic cells in the milk. Our previous study using qPCR assay also demonstrated that the genome DNAs of GM *hLZ* exist in milk samples [24]. Thus, the results here confirmed that the established KoP1 event-specific ddPCR assay could be used to quantify GM event KoP1 content in milk.

In the risk assessment of GMOs, horizontal gene transfer (HGT) of transgenic DNAs is one important concern, especially for food safety evaluation. Generally, the approaches of HGT evaluation depends on the sensitivity of the selected analytical method [28]. Due to its high sensitivity, dPCR has been applied in the evaluation of residual GM DNAs in various tissues of animals that have been fed with transgenic plant seeds or leaves. For example, Dong et al. quantified a small amount of feed-ingested *Cry1e* gene fragment from various tissues of zebrafish using ddPCR [29]. 

In the ddPCR test of goat fecal samples, no positive droplets with fluorescent signal were observed except in the fecal samples F5 and F9. The average quantified amounts of GM event KoP1 in F5 and F9 were 10.3 and 14.3 HGE per reaction with RSD values of 28.30% and 59.34%, respectively. These results indicated that most goat fecal samples do not contain GM KoP1 content. In the ddPCR assay of goat species PRLR, only three samples were quantified with positive droplets, namely, F5, F9, and F17. The quantified amounts of each sample were 11.7, 6.3, and 20.5 HGE per reaction with RSD values of 21.57%, 26.86%, and 21.06%, respectively. Meanwhile, all 18 fecal samples were also tested employing qPCR assays, and no traditional PCR amplification curves were observed from all samples. The qPCR results showed no detectable GM KoP1 event in these 18 samples (data not shown). Compared with the results of qPCR analysis, we believed that the developed ddPCR assays were more sensitive than qPCR assay. In these three samples, the quantified GM or goat amounts were almost around the LOQ of each ddPCR assay, and the repeatability of quantified amounts were slightly low (RSD < 35%) except for sample F9 in event-specific assay. The weak and unstable positive signals in these three samples might come from shed goat digestive tract cells. Some previous studies have reported that a small number of colonic epithelial cells can shed into feces, which could be utilized for potential clinical prediction of cancers [30,31]. The results indicated that HGT of transgenic DNA into feces were unlikely although weak positive signals were observed in two of the 18 samples. To further investigate whether the positive signals were coming from body cell contamination or transgene horizontal transfer, next generation sequencing methodology might be used to identify the integration of partial transgene fragments in the recipient genome.

In all five soil samples, no positive droplet was observed, indicating that no DNA fragment of GM KoP1 event existed in soil samples where transgenic goat feces were deposited. Although exfoliated digestive tract cells might be present in feces, the genomic DNAs in the cells are likely degraded rapidly following deposition in environmental soil. Concluding from the test results of fecal and soil samples, it seems that no HGT of transgenic DNA happened from the GM goat genome to the microorganisms in the digestive tract and living environment, and the developed KoP1 ddPCR assay is a useful method for the risk assessment of transgenic animal excreta for assessing environmental safety. Our results were also consistent with previous works which used PCR and qPCR to evaluate the fate of transgenic DNA in soil environment [23,32].

## 4. Conclusions

In this study, an event-specific ddPCR assay for GM goat event KoP1 was established, which targets the junction sequences between the exogenous DNA and goat genome DNA. The specificity, sensitivity, repeatability, dynamic range, accuracy, and precision of the developed ddPCR assay were evaluated to be quite good. The assay was successfully applied in quantifying transgenic DNA in practical samples, such as milk, fecal, and soil samples, for risk assessment. The results showed that the transgenic DNAs could be quantified reliably in milk samples, and no transgenic DNAs could be stably quantified in fecal and soil samples, indicating that the developed ddPCR is suitable for tracing the transgenic DNAs in risk assessment and evaluating HGT of transgene to other species and surrounding environment.

## Figures and Tables

**Figure 1 foods-11-00868-f001:**
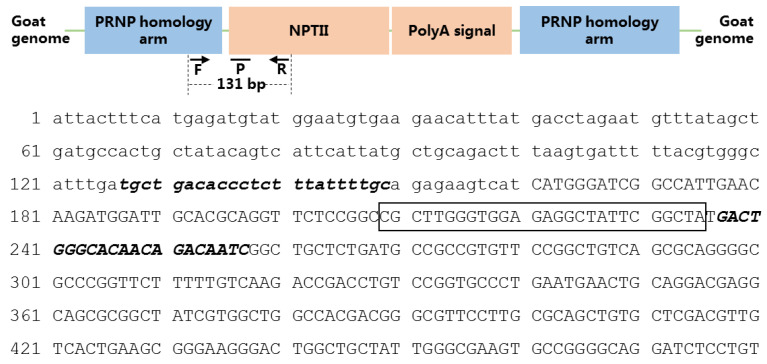
Schematic information and the location of the event-specific primers and probe in GM event KoP1. *PRNP*: prion protein gene, *NPTII*: neomycin-kanamycin phosphotransferase type II, F: forward primer, R: reverse primer, P: TaqMan probe. Lowercase letters represent the goat genomic DNA sequence, capital letters represent the *NPTII* sequence, bold italic letters represent the forward and reverse primers, letters in the frame is the TaqMan probe.

**Figure 2 foods-11-00868-f002:**
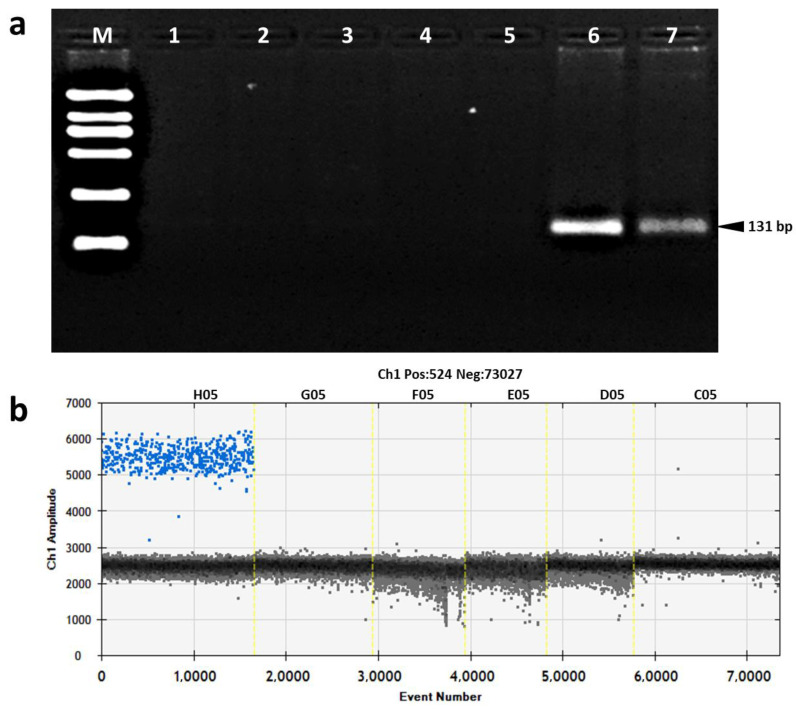
Specificity test of event-specific primer/probe set using qualitative PCR and ddPCR. (**a**) Agarose gel electrophoresis. Lane M, DL2000 DNA marker which contains six DNA fragments with lengths of 2000, 1000, 750, 500, 250, and 100 bp, respectively; lane 1: GM *hLZ* goat; lane 2: GM *hLF* goat; lane 3: GM *hSA* goat; lane 4: non GM goat; lane 5: no template control; lanes 6–7: KoP1 event. (**b**) Droplet readout of ddPCR. H05: KoP1 event; G05: GM *hLZ* goat; F05: GM *hLF* goat; E05: GM *hSA* goat; D05: non-GM goat; C05: no template control.

**Figure 3 foods-11-00868-f003:**
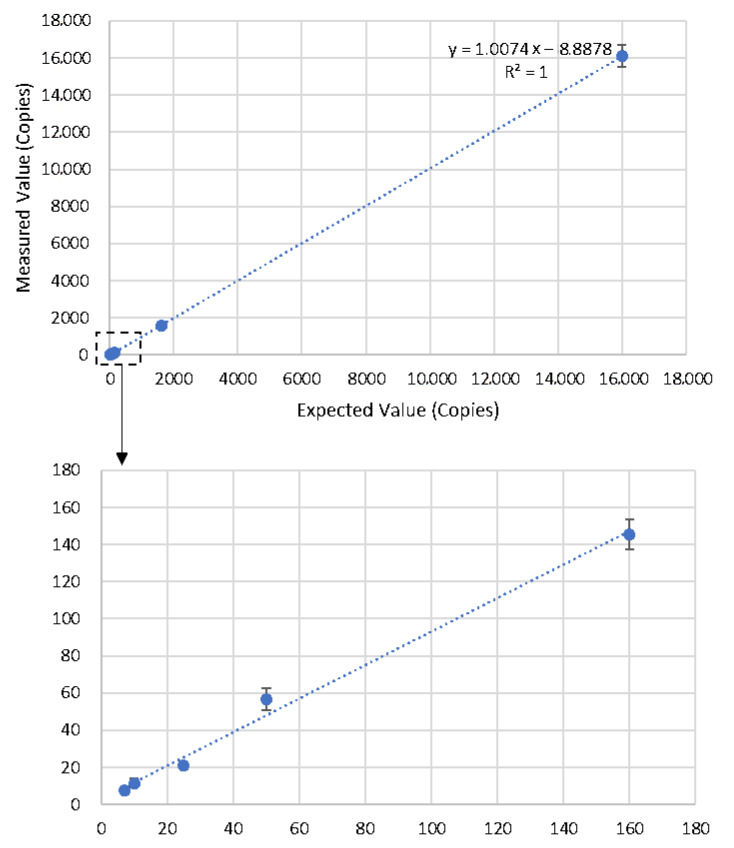
Dynamic range of KoP1 event-specific ddPCR assay.

**Table 1 foods-11-00868-t001:** List of primers and probes for ddPCR analysis.

Assay	Primer/Probe	Sequence (5′–3′)	Amplicon (bp)
KoP1	GM-Prion-F	TGCTGACACCCTCTTTATTTTGC	131
	GM-Prion-R	GATTGTCTGTTGTGCCCAGTC	
	GM-Prion-P	FAM-TAGCCGAATAGCCTCTCCACCCAAGCG-BHQ1	
*PRLR*	Goat-F	CCAACATGCCTTTAAACCCTCAA	88
	Goat-R	GGAACTGTAGCCTTCTGACTCG	
	Goat-P	FAM-TGCCTTTCCTTCCCCGCCAGTCTC-BHQ1	

**Table 2 foods-11-00868-t002:** LOD and LOQ of the KoP1 event-specific ddPCR assay.

Template Amounts (HGE/Per Reaction)	Accept Droplets				Tested HGE Copies				SD	RSD	Bias
	Rep 1	Rep 2	Rep 3		Rep 1	Rep 2	Rep 3	Mean			
4500	13,025	12,904	12,867		3940	4260	4300	4166.67	197.32	4.74%	−7%
900	15,641	14,028	14,350		956	842	816	871.33	74.47	8.55%	−3%
180	12,585	13,493	12,323		186	174	188	182.67	7.57	4.15%	1%
36	13,313	13,008	15,371		40	40	46	42.00	3.46	8.25%	17%
7.2	14,878	14,432	13,456		6.6	6.2	8.4	7.07	1.17	16.58%	−2%
1.44	13,718	13,506	13,241		3.2	3.4	3.6	3.40	0.20	5.88%	136%
0.29	/	/	/	/	/	/	/	/	/	/	/

“/” means no positive results and not detectable.

**Table 3 foods-11-00868-t003:** Dynamic range and repeatability evaluation of the event-specific ddPCR assay.

Template Amounts (HGE/per Reaction)	Tested HGE Copies				SD	RSD	Bias
	Rep1	Rep2	Rep3	Average			
16,000	16,640	16,220	15,480	16,113.33	587.31	3.64	0.71
1600	1480	1580	1640	1566.67	80.83	5.16	−2.08
160	138	144	154	145.33	8.08	5.56	−9.17
50	62	58	50	56.67	6.11	10.78	13.33
25	22	22	19	21.00	1.73	8.25	−16.00
10	14	12	9	11.67	2.52	21.57	16.67
7.2	6.9	8.2	7.8	7.63	0.67	8.72	6.02
5	2.3	8.6	3.1	4.67	3.43	73.50	−6.67
1	/	/	/	/	/	/	/

“/” means no positive results and no detectable.

**Table 4 foods-11-00868-t004:** Quantification of the GM KoP1 contents of four simulated blood DNA samples with ddPCR assays.

Sample (GM Content, %)	Assay	Accept Droplets			Tested HGE Copies				SD	RSD	GM Content	Bias
		Rep 1	Rep 2	Rep 3	Rep 1	Rep 2	Rep 3	Average				
B1 (1.7%)	KoP1	14,980	14,900	12,480	138	138	132	136.00	3.46	2.55%	1.71%	0.59%
	*PRLR*	10,643	10,877	11,500	7620	8100	8180	7966.67	302.88	3.80%		
B2 (0.85%)	KoP1	16,542	15,859	12,803	62	65	74	67.00	6.24	9.32%	0.86%	1.18%
	*PRLR*	11,854	10,364	11,925	7520	7500	8280	7766.67	444.67	5.73%		
B3 (0.5%)	KoP1	11,069	16,058	11,970	46	44	40	43.33	3.06	7.05%	0.55%	10.00%
	*PRLR*	11,854	11,925	11,500	7690	7780	7960	7810.00	137.48	1.76%		
B4 (0.25%)	KoP1	11,441	13,213	13,213	20	24	22	22.00	2.00	9.09%	0.28%	12.00%
	*PRLR*	12,764	11,642	12,850	7910	7790	8040	7913.33	125.03	1.58%		

**Table 5 foods-11-00868-t005:** Detection of practical blood, fecal, and soil samples using the developed event-specific ddPCR assay.

Sample Type	Sample Name	Animal Type	KoP1 Event-Specific Assay						Goat Species Assay					
			Tested Copies Per Reaction				SD	RSD	Tested Copies Per Reaction				SD	RSD
			Rep 1	Rep 2	Rep 3	Mean			Rep 1	Rep 2	Rep 3	Mean		
Milk	46	GM	1340	1340	1360	1346.7	11.55	0.86%	1568	1610	1520	1566.0	45.03	2.88
	323	GM	984	1056	1020	1020.0	36.00	3.53%	1010	1050	1028	1029.3	20.03	1.95
	1606	GM	910	958	890	919.3	34.95	3.80%	906	942	930	926.0	18.33	1.98
	1608	GM	820	830	848	832.7	14.19	1.70%	804	796	782	794.0	11.14	1.40
Fresh Feces	F1	GM	N	N	N	/	/	/	N	N	N	/	/	/
	F2	GM	N	N	N	/	/	/	N	N	N	/	/	/
	F3	GM	N	N	N	/	/	/	N	N	N	/	/	/
	F4	GM	N	N	N	/	/	/	N	N	N	/	/	/
	F5	GM	12	8	N	10	2.83	28.30%	14	12	9	11.7	2.52	21.57
	F6	GM	N	N	N	/	/	/	N	N	N	/	/	/
	F7	GM	N	N	N	/	/	/	N	N	N	/	/	/
	F8	GM	N	N	N	/	/	/	N	N	N	/	/	/
	F9	GM	11	24	8	14.3	8.5	59.34%	8	6.4	4.6	6.3	1.70	26.86
	F10	GM	N	N	N	/	/	/	N	N	N	/	/	/
	F11	GM	N	N	N	/	/	/	N	N	N	/	/	/
	F12	GM	N	N	N	/	/	/	N	N	N	/	/	/
	F13	GM	N	N	N	/	/	/	N	N	N	/	/	/
	F14	GM	N	N	N	/	/	/	N	N	N	/	/	/
	F15	GM	N	N	N	/	/	/	N	N	N	/	/	/
	F16	GM	N	N	N	/	/	/	N	N	N	/	/	/
	F17	Non-GM	N	N	N	/	/	/	20.8	24.6	16	20.5	4.31	21.06
	F18	Non-GM	N	N	N	/	/	/	N	N	N	/	/	/
Compost Soil	P1	GM	N	N	N	/	/	/	N	N	N	/	/	/
	P2	GM	N	N	N	/	/	/	N	N	N	/	/	/
	P3	GM	N	N	N	/	/	/	N	N	N	/	/	/
	P4	GM	N	N	N	/	/	/	N	N	N	/	/	/
	P5	GM	N	N	N	/	/	/	N	N	N	/	/	/

“N” means negative result; “/” means no data.

## Data Availability

Data is contained within the article or Appendix A.

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
