# Peer review of "Development of an Event-Specific Droplet Digital PCR Assay for Quantification and Evaluation of the Transgene DNAs in Trace Samples of GM PRNP-Knockout Goat"

_foods, 2022, doi:10.3390/foods11060868_

Round 1

Reviewer 1 Report

Please revise the manuscript according to the comments below.

Comments to the manuscript “Development of One Event-Specific Droplet Digital PCR Assay for Quantifying and Evaluating the Transgene DNAs in Trace Samples of GM PRNP-Knockout Goat”

Point 1

Title: Development of an Event-Specific Droplet Digital PCR Assay

Point 2

Modify the abstract which contains some repetitions, e.g as follows:

Furthermore, this assay was successfully applied in quantifying the goat KoP1 GM content in milk, faeces and living environmental soil samples. The test results showed that this ddPCR assay is suitable for quantifying KoP1 GM event in trace samples. has the potential to be used in the evaluation of horizontal gene transfer, assess risks of transgenic animals and their derivatives residues to the living environment. We believe that the developed ddPCR assay could be used for quantifying KoP1 GM content not only in routine lab analysis, but also in 21 has the potential to be used in the evaluation of horizontal gene transfer and the practical risk assessment of GM goat event KoP1 and its derivatives.

Point 3

Correct the citation style throughout the document, e.g. line 32: organ xenotransplantation research [1].

Point 4

Check the correct abbreviations of prion protein (PRNP or Prnp or PRP) and gene (PrP) throughout the document

Point 5

GMO screening methods can be element-specific, construct-specific or gene-specific (see JRC GMO-matrix application https://gmo-crl.jrc.ec.europa.eu/jrcgmomatrix/matrices/full). Correct the following sentences beginning in line 50 accordingly.

The DNA detection methods for GMOs are grouped into four categories according to their specificities to exogenous DNA sequences: element-specific, gene-specific, construct- specific and event-specific [8]. Normally, screening-specific and event-specific methods target the common used transgenic elements (CaMV35s promoter, NOS terminator and NptII gene, etc.) while element-specific methods detect the junction region between recipient genome and exogenous inserted DNAs. and They are mainly used in routine lab analysis of GMOs. Screening method are firstly applied to preliminarily identify whether or not the samples contain GM components, then …

Point 6,  line 64:

We previously reported an event-specific PCR method for quantifying the GM content of GM hLZ goat event [13]. – wrong citation, see [24]

Point 7, line 86

To date, only a few ddPCR assays for GM animal content in risk 86 assessment have been reported. – please give/cite examples

Point 8, line 132

The locations and sequences of event-specific primers (GM-Prion-F/-R) and probe (GM-Prion-P) are shown in Figure 1.

Point 9, line 167

… employing genomics DNA

Point 10, line 173

…corresponding

 Point 11, chapter 3.1

It was shown that the event-specific conventional PCR and ddPCR detects GM KoP1 and not the other GM goat lines or NGM goat. As the NPTII gene is widely used in plant GMOs possible cross reactions with GM crops could be relevant for testing environmental samples. As such tests are not shown in the manuscript a conclusion that the assay has high specificity seems to be too far-reaching. Please delete the word “high” or revise the chapter accordingly.

Point 12, Table 2, chapters 2.5 and 3.2

With which method were the HGEs of the genomic DNA of GM event KoP1 determined?

Are the droplet numbers for independent tests 1, 2 and 3 (e.g. at three days) mean values of three repeats each? Please clarify.

Point 13, Table 5, chapter 3.5

Correct spelling: “faeces” not “faces”

From 18 faeces samples 16 were from GM KoP1 and 2 from NGM. Please give information which of the samples are GM and which not and mention this aspect in chapter 3.5. In addition please provide further data for the faeces species of ddPCR results for the endogenous reference gene PRLR. Most likely positive KoP1 detection is due to some shed goat digestive tract cells and not to horizontal gene transfer.

Author Response

Response to the comments of referee 1

Comment 1:

This article describes the development of new ddPCR assay for Quantification of transgenic DNA in goats. Authors claimed that the assay described in the article is more sensitive and reliable than qPCR which they explained a golden standard method in this area. In order to support their claims all of the data presented in the article should be compared with qPCR data. However, no qPCR data is presented in the article.

In addition, the data acquired with the field samples do not match with author’s statements. More filed samples should be tested to get the value they claimed in the manuscript.

In summary, the manuscript should be reviewed again after major revision.

Answer: Thanks for your kind comments. We have added the qPCR analyzed results of trace samples and compared with those of ddPCR analysis in the revised version. In the trace samples, in particular for the soil and feces samples, the transgenic prion goat DNAs should be undetectable with the negative result in theory, if there is no exfoliation of intestinal epidermal cells and horizontal gene transfer. The negative results of soil and feces samples just showed that there is no horizontal gene transfer, and the sporadic positive results in feces samples might come from the exfoliation of intestinal epidermal cells.

We also compared the developed ddPCR assays with the qPCR assays in the new MS. For example, the ddPCR presents lower LOD and LOQ than qPCR. Also, the ddPCR can detect the GM and goat species DNA from three samples, while no PCR amplification curves were observed in all samples using qPCR assay, which was shown in the Figure as below. Therefore, we believe that ddPCR is more sensitive and reliable than qPCR in trace samples analysis.

Comment 2:

Tittle: -The words quantifying, and evaluation can be changed to quantification and evaluation, respectively.

Answer: Thanks for your kind comment. We have revised it in the new MS.

Comment 3:

Abstract: -Full names for the abreactions needed.

Answer: Thanks for your kind comment. We have revised it in the new MS.

Comment 4:

Materials and methods: -The instrument used for qPCR should be stated.

Answer: Thanks for your kind comment. We have added the information of PCR instrument in the new MS.

Comment 5:

Results and discussion: -Separate Results section and discussion section is recommended.

Answer: Thanks for your kind comment. We prepared the MS according to the suggested format of “Foods”.

Comment 6:

-Fig. 2a: Description about molecular weight markers is needed. There are 2 bands appeared in lane 2. The appearance of 2 bands does not match with authors opinion.

Answer: Thanks for your kind comment. We have added the details of molecular marker DL2000 in the revised version. We also replaced the Figures with one new clear version.

Comment 7:

-Line 209: Acceptance level of CV value is self-referenced. Other references are needed to support author’s statement.

Answer: Thanks for your kind comment. We have cited the original file from ENGL in the new MS.

Comment 8:

-Fig. 3: To support R2 value, data points between 2,000~14,000 are required.

Answer: Thanks for your kind comment. In order to evaluate the dynamic range of developed ddPCR assay, a total of nine DNA dilutions with the concentration of 16,000, 1600, 160, 50, 25, 10, 7.2, 5, and 1 HGE/μL were used. In general, the ddPCR is quite accurate with the range of 20000 to 1000. Therefore, we did not set a finer concentration between 16000 and 1600 HGE/μL. Our results also confirmed that the developed ddPCR assay has very good linearity regression (R2=1.000) within the range of 16000 to 7.2 copies HGE per reaction.

Comment 9:

-Lines 235-238: Reference is required to support the statement a=that the RSD values represent a good repeatability.

Answer: Thanks for your kind comment. We have added the reference in the new MS.

Comment 10:

-Table 5: The RSD values for field sample test is too large. And no data is acquired in many samples. Therefore, data do not support that the assay developed in this study is accurate is sensitive.

Answer: Thanks for your reminder. We have checked the data, and we found a statistic error in the F5 sample analysis. We have revised the data in the new version. In the feces sample of F5 and F9, the RSD values were 28.30% and 59.34%, which indicated that the repeatability and accuracy was not good enough in the samples with trace GM DNA closed to the low limit of the dynamic range. However, the results suggested us to be more careful in quantitative analysis of the trace samples.

Reviewer 2 Report

This article describes the development of new ddPCR assay for Quantification of transgenic DNA in goats. Authors claimed that the assay described in the article is more sensitive and reliable than qPCR which they explained a golden standard method in this area. In order to support their claims all of the data presented in the article should be compared with qPCR data.  However, no qPCR data is presented in the article.

In addition, the data acquired with the field samples do not match with author’s statements. More filed samples should be tested to get the value they claimed in the manuscript.

In summary, the manuscript should be reviewed again after major revision.

Manor points are as below.

Tittle:

-The words quantifying, and evaluation can be changed to quantification and evaluation, respectively.

Abstract:

-Full names for the abreactions needed.

Materials and methods:

-The instrument used for qPCR should be stated.

Results and discussion:

-Separate Results section and discussion section is recommended.

-Fig. 2a: Description about molecular weight markers is needed. There are 2 bands appeared in lane 2. The appearance of 2 bands does not match with authors opinion.

-Line 209: Acceptance level of CV value is self-referenced. Other references are needed to support author’s statement.

-Fig. 3: To support R2 value, data points between 2,000~14,000 are required.

-Lines 235-238: Reference is required to support the statement a=that the RSD values represent a good repeatability.

-Table 5: The RSD values for field sample test is too large. And no data is acquired in many samples. Therefore, data do not support that the assay developed in this study is accurate is sensitive.

Author Response

Response to the comments of referee 2

Comments to the manuscript “Development of One Event-Specific Droplet Digital PCR Assay for Quantifying and Evaluating the Transgene DNAs in Trace Samples of GM PRNP-Knockout Goat”

Point 1

Title: Development of an Event-Specific Droplet Digital PCR Assay

Answer: Thanks for your kind comment. We have revised it in the new MS.

Point 2

Modify the abstract which contains some repetitions, e.g as follows:

Furthermore, this assay was successfully applied in quantifying the goat KoP1 GM content in milk, faeces and living environmental soil samples. The test results showed that this ddPCR assay is suitable for quantifying KoP1 GM event in trace sampleshas the potential to be used in the evaluation of horizontal gene transfer, assess risks of transgenic animals and their derivatives residues to the living environment. We believe that the developed ddPCR assay could be used for quantifying KoP1 GM content not only in routine lab analysis, but also in 21 has the potential to be used in the evaluation of horizontal gene transfer and the practical risk assessment of GM goat event KoP1 and its derivatives.

Answer: Thanks for your kind comment. We have revised it in the new MS.

Point 3

Correct the citation style throughout the document, e.g. line 32: organ xenotransplantation research [1].

Answer: Thanks for your kind comment. We have revised all citations in the new MS.

Point 4

Check the correct abbreviations of prion protein (PRNP or Prnp or PRP) and gene (PrP) throughout the document

Answer: Thanks for your kind comment. We have formatted the abbreviations of prion protein and gene in the new MS.

Point 5

GMO screening methods can be element-specific, construct-specific or gene-specific (see JRC GMO-matrix application https://gmo-crl.jrc.ec.europa.eu/jrcgmomatrix/matrices/full). Correct the following sentences beginning in line 50 accordingly.

The DNA detection methods for GMOs are grouped into four categories according to their specificities to exogenous DNA sequences: element-specific, gene-specific, construct- specific and event-specific [8]. Normally, screening-specific and event-specific methods target the common used transgenic elements (CaMV35s promoter, NOS terminator and NptII gene, etc.) while element-specific methods detect the junction region between recipient genome and exogenous inserted DNAs. and They are mainly used in routine lab analysis of GMOs. Screening method are firstly applied to preliminarily identify whether or not the samples contain GM components, then …

Answer: Thanks for your kind comment. We have revised it according to you comment in the new MS.

Point 6,  line 64:

We previously reported an event-specific PCR method for quantifying the GM content of GM hLZ goat event [13]. – wrong citation, see [24]

Answer: Thanks for your kind comment. We have revised it in the new MS.

Point 7, line 86

To date, only a few ddPCR assays for GM animal content in risk assessment have been reported. – please give/cite examples

Answer: Thanks for your kind comment. We have added the examples in the new MS.

Point 8, line 132

The locations and sequences of event-specific primers (GM-Prion-F/-R) and probe (GM-Prion-P) are shown in Figure 1.

Answer: Thanks for your kind comment. We have revised it in the new MS.

Point 9, line 167

… employing genomics DNA

 Answer: Thanks for your kind comment. We have revised it in the new MS.

Point 10, line 173

…corresponding

Answer: Thanks for your kind comment. We have revised it in the new MS.

Point 11, chapter 3.1

It was shown that the event-specific conventional PCR and ddPCR detects GM KoP1 and not the other GM goat lines or NGM goat. As the NPTII gene is widely used in plant GMOs possible cross reactions with GM crops could be relevant for testing environmental samples. As such tests are not shown in the manuscript a conclusion that the assay has high specificity seems to be too far-reaching. Please delete the word “high” or revise the chapter accordingly.

Answer: Thanks for your kind comment. In our work, we developed one event-specific ddPCR assay targeting the junction region between transgene and host genome DNA, which is more specific than those targeting the transgenic elements or transgenes because on the GM KoP1 event presents the event-specific DNA sequence. However, the transgenic elements or marker genes, such as FMV35s promoter, NOS terminator, and NPTII, etc., were often introduced into multiple GM events. Therefore, the element-specific methods targeting elements or marker genes were often used to screening the GM contents before further identifying the GM event in routine lab analysis. That is the reason why we developed the event-specific ddPCR, and did not use the NPTII detection method in this study.

Point 12, Table 2, chapters 2.5 and 3.2

With which method were the HGEs of the genomic DNA of GM event KoP1 determined?

Are the droplet numbers for independent tests 1, 2 and 3 (e.g. at three days) mean values of three repeats each? Please clarify.

 Answer: Thanks for your kind comment. In the preparation of the GM goat genomic DNA dilutions, the concentration of the initial mother solution was measured using Quant-iT™ PicoGreen™ dsDNA Assay Kit, and the HGE value was determined according to the the genome size of haploid goat. In ddPCR assay, the absolute HGE copies were auto-calculated by the QuantaSoft of ddPCR machine according to the positive droplet numbers and the Poisson distribution formula.

For ddPCR analysis of each dilution or each sample, each sample was repeated three times, with 3 parallel reactions for each repeat. In table 2, the mean value of each repeat was listed, and we have added the mean value in Table 2.

 Point 13, Table 5, chapter 3.5

Correct spelling: “faeces” not “faces”

Answer: Thanks for your kind comment. We have revised it in the new MS.

Point 14

From 18 faeces samples 16 were from GM KoP1 and 2 from NGM. Please give information which of the samples are GM and which not and mention this aspect in chapter 3.5. In addition, please provide further data for the faeces species of ddPCR results for the endogenous reference gene PRLR. Most likely positive KoP1 detection is due to some shed goat digestive tract cells and not to horizontal gene transfer.

Answer: Thanks for your kind comment. We have labeled the 18 feces samples. Also, we have added the ddPCR results of endogenous reference gene PRLR for each sample in Table 5.